# An Ensemble Method for Feature Screening

**Xi Wu [1], Shifeng Xiong [2]** and **Weiyan Mu [3],***

1 National Cancer Center, National Clinical Research Center for Cancer, Cancer Hospital, Peking Union Medical College, Chinese Academy of Medical Sciences, Beijing 100021, China
2 NCMIS, KLSC, Academy of Mathematics and Systems Science, Chinese Academy of Sciences, Beijing 100190, China
3 School of Science, Beijing University of Civil Engineering and Architecture, Beijing 100044, China
* Correspondence:muweiyan@bucea.edu.cn

**Abstract:** It is known that feature selection/screening for high-dimensional nonparametric models is an important but very difficult issue. In this paper, we first point out the limitations of existing screening methods. In particular, model-free sure independence screening methods, which are defined on random predictors, may completely miss some important features in the underlying nonparametric function when the predictors follow certain distributions. To overcome these limitations, we propose an ensemble screening procedure for nonparametric models. It elaborately combines several existing screening methods and outputs a result close to the best one of these methods. Numerical examples indicate that the proposed method is very competitive and has satisfactory performance even when existing methods fail.

**Keywords:** classification; nonparametric regression; R-square; sparsity; sure independent screening; variable selection

**MSC:** 62G08; 62H30

## 1. Introduction

In this paper, we study the feature screening problem for nonparametric models when the number of predictors, $p$, is larger than the number of observations, $p$. The nonparametric models we focus on include nonparametric interpolation, regression, and classification models. Interpolation is commonly used in modeling computer simulation experiments [1] and spatial data [2]. Nonparametric regression and classification are basic issues in statistics and machine learning [3]. High-dimensional problems are commonly encountered in these areas.

For linear regression models with $p > n$, Fan and Lv proposed the sure independence screening (SIS) method to screen the subsets of important/active features/predictors/variables [4]. This method was extended to generalized linear models [5], nonparametric additive models [6], and varying coefficient models [7]. Many model-free SIS-type methods were also provided in the literature, see, e.g., [8–12], and they can be used for the aforementioned high-dimensional nonparametric models. These SIS-type methods consider random predictors and screen active predictors by ranking some independence indices, which describe the independence between each predictor and the response.

For nonparametric interpolation models with $p > n$, Li, Chen, and Xiong proposed the linear screening method [13]. Their idea stems from the method of selecting important variables via the main effects in the experimental design literature [14]. This method adopts linear regression to model the data from nonlinear models and uses the $\ell_0$-screening [15,16] or $\ell_1$-screening principle [17] for the linear model to screen the active variables. Li, Chen, and Xiong proved that the linear screening method is asymptotically valid under some

conditions [13]. Numerical simulations show that this method performs better than SIS-type methods in most cases for interpolation models.

Note that the linear screening method can also be used for other nonparametric problems, including regression and classification. In this paper, we first discuss the two classes of nonparametric screening methods, the SIS-type and linear screening methods, and then point out their own limitations. For the SIS-type methods, we provide an example where the regression function depends on a variable while the random version of this variable is independent of the response. Therefore, any independence index-based SIS-type method cannot screen this important variable of the regression model. For the linear screening method, we show that when the function is not close to a linear norm and the predictors are correlated, it may perform very poorly. To overcome these limitations, we propose an ensemble procedure to combine the two classes of methods. This procedure conducts the linear screening methods with linear and quadratic basis functions first and uses $R^2$ to evaluate their performance. If the values of $R^2$ are low, then we further conduct an SIS-type method and output the result of this SIS-type method. Otherwise, we output the subset selected by the linear screening method corresponding to the higher $R^2$. The ensemble method is very simple and easy to implement. Simulations and several real examples show that it is very competitive and has satisfactory performance even when the SIS-type methods or the linear screening methods fail.

This article is organized as follows. Section 2 gives a general description of sparse function models. Section 3 discusses the limitations of the SIS-type and linear screening methods. Section 4 presents the proposed ensemble method. Section 5 illustrates the methods with several datasets. We conclude the paper with some concluding remarks in Section 6. More simulation results and technical proofs are given in the Supplementary Materials.

## 2. Sparse Function Models

This section discusses some basic characteristics of sparse function models in a rigorous mathematical manner. We will emphasize that the sparsity property of a function does not depend on the distributions of the covariates/input variables.

Suppose the unknown function of interest is $f(\mathbf{x})$, where the features/variables $\mathbf{x} = (x_1, \ldots, x_p)' \in \mathcal{I}^p$, $f$ is square integrable, i.e., $f \in L^2(\mathcal{I}^p)$, $\mathcal{I} \subset \mathbb{R}$ is an interval, and 'denotes the transpose. Without loss of the generality, let $\mathcal{I} = (0,1)$, and our discussion can straightforwardly be modified to accommodate general $\mathcal{I}$ such as $(0, \infty)$ or $\mathbb{R}$. To make the following notation clear, let $\mathcal{I}_j = (0,1)$ denote the range of $x_j$ for $j = 1, \ldots, p$.

We need further definitions and notation. Let $\mathbb{Z}_p = \{1, \ldots, p\}$. For a set $\mathcal{A}$, $|\mathcal{A}|$ denotes its cardinality. For $\mathbf{x} \in \mathbb{R}^p$ and $\mathcal{A} \subset \mathbb{Z}_p$, let $\mathbf{x}_\mathcal{A}$ denote the sub-vector corresponding to $\mathcal{A}$ and let $\mathcal{I}^\mathcal{A} = \prod_{j \in \mathcal{A}} \mathcal{I}_j$. Note that $L^2(\mathcal{I}^p)$ is a closed Hilbert space with the inner product $< f, g > = \int_{\mathcal{I}^p} f(\mathbf{x}) g(\mathbf{x}) d\mathbf{x}$ for $f, g \in L^2(\mathcal{I}^p)$. We write $f = g$ in the sense of the norm in $L^2(\mathcal{I}^p)$, i.e., $\int_{\mathcal{I}^p} [f(\mathbf{x}) - g(\mathbf{x})]^2 d\mathbf{x} = 0$. For $\mathcal{A} \subset \mathbb{Z}_p$, define

$$\mathcal{L}(\mathcal{A}) = \left\{ f \in L^2(\mathcal{I}^p) : \ f(\mathbf{x}) \text{ does not depend on } \mathbf{x}_{\mathbb{Z}_p \setminus \mathcal{A}} \text{ for all } \mathbf{x} \in \mathcal{I}^p \right\},$$

which is a closed subspace of $L^2(\mathcal{I}^p)$. It can be seen that $\mathcal{L}(\mathcal{A})$ and $L^2(\mathcal{I}^\mathcal{A})$ are isomorphic. Define $f_\mathcal{A}$ to be the projection of $f$ onto $L^2(\mathcal{I}^\mathcal{A})$, i.e.,

$$f_\mathcal{A}(\mathbf{x}_\mathcal{A}) = \arg \min_{g \in L^2(\mathcal{I}^\mathcal{A})} \int_{\mathcal{I}^p} [f(\mathbf{x}) - g(\mathbf{x}_\mathcal{A})]^2 d\mathbf{x}.$$

We know that $f_\mathcal{A}$ is the unique projection in the sense of the norm in $L^2(\mathcal{I}^p)$ [18]. We make the following sparsity assumptions. Denote $\mathcal{A}_0 = \{1, \ldots, p_0\}$ with $p_0 < p$.

**Assumption 1.** $f = f_{\mathcal{A}_0}$.

**Assumption 2.** *For each $\mathcal{A} \subsetneq \mathcal{A}_0$, $f_{\mathcal{A}_0} \neq f_\mathcal{A}$.*

**Proposition 1.** *Under Assumptions 1 and 2, if there exists $\mathcal{A} \subset \mathbb{Z}_p$ with $|\mathcal{A}| \leqslant p_0$ such that $f = f_{\mathcal{A}}$, then $\mathcal{A} = \mathcal{A}_0$.*

Proposition 1 shows the uniqueness of $\mathcal{A}_0$. We call $\mathcal{A}_0$ the true subset of important/active variables for the nonparametric model $f$. The goal of screening is to find a subset $\mathcal{A}^* \supset \mathcal{A}_0$ with $|\mathcal{A}^*| = M < n$. A suggestion for selecting $M$ is $M = [n/\log(n)]$ [4], where $[\cdot]$ denotes the floor function.

Consider the situations where the variables $\mathbf{x}$ are random with a probability measure $P$ on $\mathcal{I}^p$.

**Assumption 3.** *$P$ has a density $\psi$ with respect to the Lebesgue measure, and $\psi$ is positive almost everywhere.*

Assumption 3 implies that $P$ is not degenerate. Let

$$L_P^2(\mathcal{I}^p) = \left\{ f \text{ defined on } \mathcal{I}^p \text{ is measurable} : \int_{\mathcal{I}^p} |f(\mathbf{x})| \psi(\mathbf{x}) d\mathbf{x} < \infty \right\}.$$

Similarly define $L_P^2(\mathcal{I}^{\mathcal{A}})$ for $\mathcal{A} \subset \mathbb{Z}^p$. For $f, g \in L^2(\mathcal{I}^p)$, denote $f =_P g$ if $f$ and $g$ are identical in the sense of the norm in $L_P^2(\mathcal{I}^p)$, i.e., $\int_{\mathcal{I}^p} [f(\mathbf{x}) - g(\mathbf{x})]^2 \psi(\mathbf{x}) d\mathbf{x} = 0$. Define $f_{\mathcal{A},P}$ to be the projection of $f \in L_P^2(\mathcal{I}^p)$ onto $L_P^2(\mathcal{I}^{\mathcal{A}})$, i.e.,

$$f_{\mathcal{A},P}(\mathbf{x}_{\mathcal{A}}) = \arg \min_{g \in L_P^2(\mathcal{I}^{\mathcal{A}})} \int_{\mathcal{I}^p} [f(\mathbf{x}) - g(\mathbf{x}_{\mathcal{A}})]^2 \psi(\mathbf{x}) d\mathbf{x}.$$

We know that $f_{\mathcal{A},P}$ is the unique projection in the sense of the norm in $L_P^2(\mathcal{I}^p)$. In general, $f_{\mathcal{A},P}$ will vary as $P$ varies for fixed $f$ and $\mathcal{A}$. However, we have the following proposition showing the invariance property of sparsity.

**Proposition 2.** *Under Assumption 3, for $f \in L^2(\mathcal{I}^p) \cap L_P^2(\mathcal{I}^p)$, the two following statements are equivalent:*

*(i)　$f = f_{\mathcal{A}_0}$ and $f_{\mathcal{A}_0} \neq f_{\mathcal{A}}$ for each $\mathcal{A} \subsetneq \mathcal{A}_0$;*
*(ii)　$f =_P f_{\mathcal{A}_0,P}$ and $f_{\mathcal{A}_0,P} \neq_P f_{\mathcal{A},P}$ for each $\mathcal{A} \subsetneq \mathcal{A}_0$.*

Proposition 2 indicates that the true subset $\mathcal{A}_0$ does not rely on the distribution $P$ of $\mathbf{x}$, and that the sparsity is an essential property that only depends on $f$ itself. In other words, the true subset $\mathcal{A}_0$ uniquely exists and can be identified in theory no matter what distribution the predictors follow. The above results can be extended to the cases where $\mathbf{x}$ has some discrete components by making assumptions on $P$ accordingly.

A selection principle $\mathcal{A}^* \subset \mathbb{Z}_p$ depends on the distribution $P$ of the variables $\mathbf{x}$, i.e., $\mathcal{A}^* = \mathcal{A}^*(P)$. An ideal selection principle should satisfy the selection consistency property $\mathcal{A}^*(P) = \mathcal{A}_0$ for all reasonable $P$, where $\mathcal{A}_0$ is free of $P$ by Proposition 2. For the cases of $p > n$, the selection consistency property is usually relaxed as the sure screening property $\mathcal{A}^*(P) \supset \mathcal{A}_0$. Based on the sample version $P_n$ of $P$, the sure screening property requires that the probability of $\mathcal{A}^*(P_n) \supset \mathcal{A}_0$ should tend to one as $n \to \infty$ [4]. However, for nonparametric function $f$, it is difficult to find an ideal selection principle. We will show that there exist commonly-encountered $P$'s that cause popular screening methods do not have the selection consistency property in the next section.

## 3. Discussion on Existing Methods

For a high-dimensional sparse function $f$ discussed in the previous section, the feature screening problem is to find an $M$-subset of these $p$ variables that can include all the $p_0$ active variables based on data of sample size $n$ $(<p)$, where $M$ $(<n)$ is a pre-specified number.

### 3.1. SIS-Type Methods

The SIS-type methods consider the situations with random predictors (covariates). Let $\mathbf{X} = (X_1, \ldots, X_p)'$ follow a probability measure $P$. The SIS-type methods use the selection principle

$$\mathcal{A}^*_{\text{SIS}}(P) = \{j \in \mathbb{Z}_p : X_j \text{ is not independent of } Y\}, \tag{1}$$

where $Y$ is the corresponding random response depending on the underlying function $f$. For instance, we review Li, Zhong, and Zhu [9]'s method as follows.

Li, Zhong, and Zhu used the distance correlation (DC) to screen the important variables [9]. For two random variables $U$ and $V$, the DC between them is denoted by $\text{dcorr}(U, V)$. Székely, Rizzo, and Bakirov showed that $\text{dcorr}(U, V) = 0$ if and only if $U$ and $V$ are independent [19] and that $\text{dcorr}(U, V)$ can be computed by

$$\text{dcorr}(U, V) = \frac{\text{dcov}(U, V)}{\sqrt{\text{dcov}(U, U)\text{dcov}(V, V)}},$$

where $\text{dcov}^2(U, V) = S_1 + S_2 - 2S_3$, $S_1 = E\{|U - \tilde{U}||V - \tilde{V}|\}$, $S_2 = E\{|U - \tilde{U}|\}E\{|V - \tilde{V}|\}$, $S_3 = E\{E(|U - \tilde{U}| \mid U)E(|V - \tilde{V}| \mid V)\}$, and $(\tilde{U}, \tilde{V})$ is an independent copy of $(U, V)$.

Suppose that $\{(U_i, V_i), i = 1, \ldots, n\}$ is a random sample from the population $(U, V)$. The DC $\text{dcorr}(U, V)$ can be estimated by

$$\widehat{\text{dcorr}}(U, V) = \frac{\widehat{\text{dcov}}(U, V)}{\sqrt{\widehat{\text{dcov}}(U, U)\widehat{\text{dcov}}(V, V)}}, \tag{2}$$

where $\widehat{\text{dcov}}^2(U, V) = \hat{S}_1 + \hat{S}_2 - 2\hat{S}_3$, $\hat{S}_1 = \sum_{i=1}^n \sum_{j=1}^n |U_i - U_j||V_i - V_j|/n^2$, $\hat{S}_2 = \left(\sum_{i=1}^n \sum_{j=1}^n |U_i - U_j|/n^2\right)\left(\sum_{i=1}^n \sum_{j=1}^n |V_i - V_j|/n^2\right)$, $\hat{S}_3 = \sum_{i=1}^n \sum_{j=1}^n \sum_{k=1}^n |U_i - U_k||V_j - V_k|/n^3$.

For $p$ random predictors $X_1, \ldots, X_p$ and the response $Y$, we have the sample $\{(\mathbf{X}_i, Y_i), i = 1, \ldots, n\}$, where $\mathbf{X}_i = (X_{i1}, \ldots, X_{ip})'$. The DC-SIS method first computes $\widehat{\text{dcorr}}(X_j, Y)$ for $j = 1, \ldots, p$ by (2), and then screens the subset $\mathcal{A}^*$ with $M$ variables corresponding to the $M$ largest values among $\left\{\left|\widehat{\text{dcorr}}(X_j, y)\right|\right\}_{j=1,\ldots,p}$.

We can see from (1) that the SIS-type methods greatly depend on the distribution of $\mathbf{X}$. As discussed in Section 2, the sparsity property of a function does not depend on the distributions of the covariates/input variables. The following example shows that the dependency-based selection principle does not have the selection consistency property.

**Example 1.** *Let* $f(x_1, x_2) = x_1$ *and* $Y = f(X_1, X_2)$. *If* $X_1$ *is not independent of* $X_2$, *then* $\mathcal{A}^*_{\text{SIS}}(P) = \{1, 2\} \neq \mathcal{A}_0 = \{1\}$, *where* $P$ *is the joint distribution of* $(X_1, X_2)'$.

In the above example, we can say that $\mathcal{A}^*_{\text{SIS}}(P)$ possesses at least the sure screening property $\mathcal{A}^*_{\text{SIS}}(P) \supset \mathcal{A}_0$. However, we have the following fact, which can be shown by Example 2.

**Fact 1.** *The SIS-type methods may lack the screening property, i.e., some important features may be missed.*

**Example 2.** *Let*

$$f(x_1, x_2) = \Phi^{-1}(x_1) + 2\Phi^{-1}(x_2), \ (x_1, x_2)' \in (0, 1)^2, \tag{3}$$

*in a regression model*

$$y = f(x_1, x_2) + \varepsilon$$

with $\varepsilon \sim N(0,1)$, where $\Phi$ denotes the cumulative distribution function of the standard normal distribution. Consider the random $(X_1, X_2)'$, independent of $\varepsilon$, generated as follows. First generate $X_1 \sim U(0,1)$, the uniform distribution on $(0,1)$. Then generate $W$, given $X_1$, from $N(-\Phi^{-1}(X_1)/2, 3/4)$, and let $X_2 = \Phi(W)$. We have

$$
\begin{pmatrix} \Phi^{-1}(X_1) \\ \Phi^{-1}(X_2) \end{pmatrix} \sim N\left( \begin{pmatrix} 0 \\ 0 \end{pmatrix}, \begin{pmatrix} 1 & -1/2 \\ -1/2 & 1 \end{pmatrix} \right),
$$

which implies $\operatorname{cov}(Y, \Phi^{-1}(X_1)) = \operatorname{cov}(\Phi^{-1}(X_1) + 2\Phi^{-1}(X_2) + \varepsilon, \Phi^{-1}(X_1)) = 0$. Therefore, the response $Y$ is independent of $X_1$. We cannot find the important variable $x_1$ in $f$ in (3) via dependency.

The case provided in Example 2 seems to be extreme since it depends on special $f$ and $P$. In fact, for cases that are not so extreme, the performance of the SIS-type methods may be unsatisfactory. This is because they are all marginal methods and may miss the important variables with weak marginal effects in finite-sample cases. Recently Azadkia and Chatterjee proposed a model-free variable selection method via a measure of conditional dependence [20]. Their method possesses appealing theoretical properties. However, our numerical experience indicates that its finite-sample performance is not good for large $p$, small $n$ problems. SIS-type methods such as DC-SIS outperform it in many cases. Some examples are given in our simulations in the Supplementary Materials.

### 3.2. Linear Screening Methods

Li, Chen, and Xiong proposed the linear screening method for nonparametric interpolation models [13]. Unlike the marginal SIS-type methods, this method is based on linear approximations of the underlying nonparametric function. Linear screening uses a linear regression model with specific basis functions to fit the interpolation data and applies the $\ell_0$ or $\ell_1$ methods for linear regression models to screen the active variables in the nonparametric function $f$. Li, Chen, and Xiong showed that under some mild conditions, the set of active variables in $f$ is identical to that in the linear projection of $f$ [13]. Therefore, the linear screening method is valid for many cases. Since methods based on dependence and conditional dependence, such as SIS-type methods, suffer from very slow convergence in high dimensions, linear screening can be viewed as an alternative for finite-sample cases.

Li, Chen, and Xiong [13] only discussed the case where the predictors $\mathbf{x} = \mathbf{X} \sim U(0,1)^p$. This subsection further studies the linear screening methods with a general distribution $P$ of $\mathbf{x}$, where $P$ satisfies Assumption 3.

For a basis function $b \in L^2_P(\mathcal{I})$ that is not a constant, define

$$
\mathcal{L}_b = \left\{ g(\mathbf{x}) = \phi_0 + \sum_{j=1}^p \phi_j b(x_j) \text{ for } \mathbf{x} \in \mathcal{I}^p : \phi_0, \phi_1, \ldots, \phi_p \in \mathbb{R} \right\},
$$

which is a closed subspace of $L^2_P(\mathcal{I}^p)$. Consider the projection of $f \in L^2(\mathcal{I}^p) \cap L^2_P(\mathcal{I}^p)$ onto $\mathcal{L}_b$ with respect to the norm of $L^2_P(\mathcal{I}^p)$,

$$
\beta_0(P) + \sum_{j=1}^p \beta_j(P) b(x_j) = \arg\min_{g \in \mathcal{L}_b} \int_{\mathcal{I}^p} [f(\mathbf{x}) - g(\mathbf{x})]^2 \psi(\mathbf{x}) d\mathbf{x}. \tag{4}
$$

Denote

$$
\mathbf{u} = \left( \int_{\mathcal{I}^p} b(x_1)\psi(\mathbf{x}) d\mathbf{x}, \ldots, \int_{\mathcal{I}^p} b(x_p)\psi(\mathbf{x}) d\mathbf{x} \right)',
$$

$$
\mathbf{v} = \left( \int_{\mathcal{I}^p} b(x_1)f(\mathbf{x})\psi(\mathbf{x}) d\mathbf{x}, \ldots, \int_{\mathcal{I}^p} b(x_p)f(\mathbf{x})\psi(\mathbf{x}) d\mathbf{x} \right)',
$$

$$
\mathbf{\Sigma} = \left( \int_{\mathcal{I}^p} b(x_i)b(x_j)\psi(\mathbf{x}) d\mathbf{x} \right)_{i,j=1,\ldots,p}.
$$

Note that the matrix $\begin{pmatrix} 1 & \mathbf{u}' \\ \mathbf{u} & \boldsymbol{\Sigma} \end{pmatrix}$ is invertible (see the proof of Lemma B.3 in the Supplementary Materials). By solving the quadratic optimization problem in (4), we have

$$\begin{pmatrix} \beta_0(P) \\ \boldsymbol{\beta}(P) \end{pmatrix} = \begin{pmatrix} 1 & \mathbf{u}' \\ \mathbf{u} & \boldsymbol{\Sigma} \end{pmatrix}^{-1} \begin{pmatrix} \int_{\mathcal{I}^p} f(\mathbf{x})\psi(\mathbf{x})d\mathbf{x} \\ \mathbf{v} \end{pmatrix}, \tag{5}$$

where $\boldsymbol{\beta}(P) = (\beta_1(P), \ldots, \beta_p(P))'$. The selection principle of linear screening with basis $b$ is

$$\mathcal{A}_{\mathrm{LS}}^*(P) = \{j \in \mathbb{Z}_p : \beta_j(P) \neq 0\}.$$

The following proposition is obvious.

**Proposition 3.** *Under Assumptions 1 and 2, if $f \in \mathcal{L}_b$, then $\mathcal{A}_{\mathrm{LS}}^*(P) = \mathcal{A}_0$ for all $P$ satisfying Assumption 3.*

Proposition 3 provides an ideal selection principle. However, the condition of $f$ in the proposition is very strong. For general $f$, the following assumptions are needed to make the linear screening method have the selection consistency.

**Assumption 4.** $\psi(\mathbf{x}) = \prod_{j=1}^{p} \varphi(x_j)$ *for all $\mathbf{x} \in \mathcal{I}^p$, where $\varphi$ is a density function on $\mathcal{I}$.*

**Assumption 5.** *For $j = 1, \ldots, p_0$, $\int_{\mathcal{I}^p} b(x_j) f_{\mathcal{A}_0}(\mathbf{x}_{\mathcal{A}_0})\psi(\mathbf{x})d\mathbf{x} \neq \int_{\mathcal{I}} b(x)\varphi(x)dx \int_{\mathcal{I}^p} f_{\mathcal{A}_0}(\mathbf{x}_{\mathcal{A}_0})\psi(\mathbf{x})d\mathbf{x}$.*

**Proposition 4.** *Under Assumptions 1–5, $\mathcal{A}_{\mathrm{LS}}^*(P) = \mathcal{A}_0$.*

The following example shows that, without Assumption 4, Proposition 4 may not hold.

**Example 3.** *Let $f(x_1, x_2) = x_1^2$ and $b(x) = x$. For $X_1$ and $X_2$ both generated from $U(0,1)$, by (5), some algebra shows that $\beta_2(P) = 0$ if and only if $E(X_1 X_2) = E(X_1)E(X_2) = 1/4$, where $P$ is the joint distribution of $(X_1, X_2)'$. Therefore, if $X_1$ and $X_2$ are correlated, then $\beta_2(P) \neq 0$, which implies $\mathcal{A}_{\mathrm{LS}}^*(P) \neq \mathcal{A}_0$.*

By Lemma B.3 in the Supplementary Materials, under Assumptions 1–4, Assumption 5 is the sufficient and necessary condition of $\mathcal{A}_{\mathrm{LS}}^*(P) = \mathcal{A}_0$. We give an example when Assumption 5 does not hold.

**Example 4.** *Let*

$$f(\mathbf{x}) = \prod_{j=1}^{p_0} (2x_j - 1) \tag{6}$$

*with $p_0 > 1$. Suppose $P$ is the probability measure of $U(0,1)^p$, which satisfies Assumption 4. For $j = 1, \ldots, p_0$, $\int_{\mathcal{I}^p} b(x_j) f_{\mathcal{A}_0}(\mathbf{x}_{\mathcal{A}_0})\psi(\mathbf{x})d\mathbf{x} = \int_{\mathcal{I}^{p_0}} b(x_j)(2x_j - 1) \prod_{k \neq j}(2x_k - 1)dx_1 \cdots dx_{p_0} = \int_{\mathcal{I}} b(x_j)(2x_j - 1)dx_j \prod_{k \neq j} \int_{\mathcal{I}}(2x_k - 1)dx_k = 0$, and $\int_{\mathcal{I}^p} f_{\mathcal{A}_0}(\mathbf{x})\psi(\mathbf{x})d\mathbf{x} = \prod_{k=1}^{p_0} \int_{\mathcal{I}}(2x_k - 1)dx_k = 0$. Therefore, Assumption 5 does not hold for any basis function $b$. In fact, the linear screening method selects the variables whose main effects exist, while there are only interactions in Function (6).*

The above two examples actually show the following fact.

**Fact 2.** *The linear screening methods may lack the screening property when Assumption 4 or Assumption 5 does not hold.*

Generally speaking, Assumption 4 is not easy to hold but is relatively easy to verify. When Assumption 4 holds, Assumption 5 holds for most cases except for some extreme cases such as Example 4. This is because inequality is usually easier to hold than equality. For example, if $f$ is assumed to be a realization from a Gaussian process, then Assumption 5 holds with a probability of one.

From the discussion in this subsection, we draw the overall conclusions. (a) By Proposition 3, if $f$ is close to $\mathcal{L}_b$, then the linear screening method with basis $b$ performs well for most $P$. (b) By Proposition 4, if $X_1, \ldots, X_p$ are identically and independently distributed, then the linear screening method is good for many cases. (c) If neither (a) nor (b) holds, then linear screening may be poor.

## 4. Ensemble Screening

In the previous section, we have pointed out the limitations of the SIS-type and linear screening methods for nonparametric models. To overcome these limitations, here we propose an ensemble procedure that combines the two classes of methods. First consider the linear screening methods, and we know that they perform very well when $f$ is close to its linear approximation. We can use some fitting index to check whether $f$ is close to its linear approximation or not. If the linear screening method with a candidate $b$ passes the check, then we select it to output the screening result; otherwise, we consider the SIS-type methods. The fitting index is selected as the R-square

$$R^2 = \frac{\sum_{i=1}^n (\hat{Y}_i - \bar{Y})^2}{\sum_{i=1}^n (Y_i - \bar{Y})^2}, \tag{7}$$

where $\bar{Y} = \sum_{i=1}^n Y_i / n$ and $\hat{Y}_i$ is the $i$th prediction value of the response from the linear screening method.

Specifically, for interpolation and regression problems, we use the lasso method [17] for the linear model to screen the important feature variables. The ten-fold cross-validation is used to specify the tuning parameter in the lasso. If the number of selected features is larger than $M$, we only keep the features with the largest $M$ absolute values of coefficients. Let $\mathcal{A}_{\text{LS}}^*$ denote the set of selected features. Then $(\hat{Y}_1, \ldots, \hat{Y}_n)' = \hat{\beta}_0 \mathbf{1}_n + b(\mathbf{X})_{\mathcal{A}_{\text{LS}}^*} \hat{\boldsymbol{\beta}}$ used in (7), where $\mathbf{1}_n$ denotes the $n$-vector whose components are all 1's, $b(\mathbf{X})_{\mathcal{A}_{\text{LS}}^*}$ is the sub-matrix of $b(\mathbf{X}) = (b(X_{ij}))_{i=1,\ldots,n,j=1,\ldots,p}$ corresponding to $\mathcal{A}_{\text{LS}}^*$, and $(\hat{\beta}_0, \hat{\boldsymbol{\beta}}')'$ is the least squares estimator under the selected sub-model $\mathcal{A}_{\text{LS}}^*$.

For classification problems with $y \in \{0, 1\}$, we use the lasso method for the linear logistic model to screen the important features. Then $(\hat{Y}_1, \ldots, \hat{Y}_n)' = \text{logit}\left(\hat{\beta}_0 \mathbf{1}_n + b(\mathbf{X})_{\mathcal{A}_{\text{LS}}^*} \hat{\boldsymbol{\beta}}\right)$ used in (7), where $(\hat{\beta}_0, \hat{\boldsymbol{\beta}}')'$ is the maximum likelihood estimator under the selected sub-model $\mathcal{A}_{\text{LS}}^*$ and $\text{logit}(z_1, \ldots, z_n) = \left(\exp(z_1)/[1 + \exp(z_1)], \ldots, \exp(z_n)/[1 + \exp(z_n)]\right)'$. A number of definitions of R-square for logistic regression, including that in (7), were compared in [21]. Here we recommend using (7), which is consistent with the regression cases, by our numerical experience.

The detailed steps of the ensemble screening procedure are described as follows. Let $\delta \in (0, 1)$ be a threshold value.

**Step 1:** Conduct the linear screening method with the linear basis $b(x) = x$. Obtain the subset $\mathcal{A}_{\text{LSLB}}^*$ and the corresponding $R_{\text{LSLB}}^2$.

**Step 2:** Conduct the linear screening method with the quadratic basis $b(x) = x^2$. Obtain the subset $\mathcal{A}_{\text{LSQB}}^*$ and the corresponding $R_{\text{LSQB}}^2$.

**Step 3:** If $\max(R_{\text{LSLB}}^2, R_{\text{LSQB}}^2) < \delta$, then conduct DC-SIS, and output $\mathcal{A}^* = \mathcal{A}_{\text{DC-SIS}}^*$; if $R_{\text{LSLB}}^2 \geqslant \delta$ and $R_{\text{LSLB}}^2 \geqslant R_{\text{LSQB}}^2$, then output $\mathcal{A}^* = \mathcal{A}_{\text{LSLB}}^*$ (if the size of $\mathcal{A}_{\text{LSLB}}^*$, denoted by $|\mathcal{A}_{\text{LSLB}}^*|$, is less than $M$, then add the first $M - |\mathcal{A}_{\text{LSLB}}^*|$ variables selected by DC-SIS to $\mathcal{A}^*$);

if $R_{\mathrm{LSQB}}^2 \geqslant \delta$ and $R_{\mathrm{LSQB}}^2 > R_{\mathrm{LSLB}}^2$, then output $\mathcal{A}^* = \mathcal{A}_{\mathrm{LSQB}}^*$ (if the size of $\mathcal{A}_{\mathrm{LSQB}}^*$ is less than $M$, then add the first $M - |\mathcal{A}_{\mathrm{LSQB}}^*|$ variables selected by DC-SIS to $\mathcal{A}^*$).

There is no feasible method to find the optimal basis function in the linear screening methods [13]; see also Section 3.2. Here we only consider the two basic linear and quadratic basis functions. The linear basis function corresponds to the method of selecting important variables via the main linear effects. This basis is commonly used with two-level supersaturated designs [14,22] when the predictors can be designed. The linear screening method with the quadratic basis selects important variables via the main quadratic effects and is relevant to three-level supersaturated designs [23]. Unlike the two basis functions, other basis functions do not have clear interpretations. We also try other basis functions, including high-order polynomials and some random functions. Our numerical experience shows that more than two basis functions do not lead to significant improvement. In fact, the linear screening method with only the linear basis has much better performance than SIS-type methods in most cases under interpolation models [13]. From a practical point of view, for a new dataset, at first, one usually uses a linear model (with the linear basis) to fit it. The use of the linear basis in the proposed method is consistent with this manner.

Furthermore, we try a number of feasible implementations in the linear screening method. If we use different basis functions for different variables, similar to the non-parametric additive model or use parametric interaction models [24] to conduct variable screening, then there are too many basis functions ($\gg p$) in the linear model, and this will lead to poor screening performance since $p$ itself is large. Therefore, these more complex models are not recommended in linear screening methods.

Note that SIS-type methods are all marginal methods based on independence ranking and have similar performance in most cases. Among them, DC-SIS seems to have good performance for situations where linear screening fails, in our experience. In Step 3, we choose DC-SIS in our method.

The proposed ensemble method stems from some natural ideas. For high-dimensional settings, the data are not enough to build a complex model, and we should first try linear models. If the linear model fits the data well, we use it to screen active features. Otherwise, we consider an SIS-type method, which acts as the "last" method because of its slow convergence in high-dimensional nonparametric settings. We also consider other ensemble strategies combining the two classes of screening methods, including the method that takes a union set of the results from the two classes of methods, the method that uses other criteria such as the prediction performance instead of R-square, and others. Our numerical comparisons indicate that the proposed ensemble method outperforms these strategies.

After screening the active features, an important work is the post-selection inference, including forming valid confidence intervals for the selected coefficients and testing whether all relevant variables have been included in the model [25]. For nonparametric settings, the post-selection inference problem is quite challenging. It is a valuable topic to develop effective post-selection inference methods that can be applied to the proposed ensemble procedure in the future.

## 5. Numerical Examples

In this section, we apply the proposed screening method to analyze several simulated and real datasets. The real datasets are all from the UC Irvine Machine Learning Repository. More numerical examples can be found in the Supplementary Materials.

### 5.1. Simulations for Interpolation

This subsection considers the following test functions,

$$(\text{I}) \ f(\mathbf{x}) = \sum_{j=1}^{p_0} j x_j^2,$$

$$(\text{II}) \ f(\mathbf{x}) = -20 \exp\left(-\frac{1}{5}\sqrt{\frac{1}{p_0}\sum_{j=1}^{p_0} x_j^2}\right) - \exp\left(\frac{1}{p_0}\sum_{j=1}^{p_0}\cos(2\pi x_j)\right) + 20 + \exp(1),$$

$$(\text{III}) \ f(\mathbf{x}) = \left(\sum_{j=1}^{p_0} |x_j|\right)\exp\left[-\sum_{j=1}^{p_0}\sin\left(x_j^2\right)\right],$$

$$(\text{IV}) \ f(\mathbf{x}) = \sum_{j=1}^{p_0} x_j^2 + \left(\sum_{j=1}^{p_0} j x_j/2\right)^2 + \left(\sum_{j=1}^{p_0} j x_j/2\right)^4,$$

$$(\text{V}) \ f(\mathbf{x}) = \prod_{j=1}^{p_0}(2x_j - 1).$$

Function (I) is known as the weighted sphere model, Function (II) is Ackley's model, Function (III) is Yang's model, and Function (IV) is Zakharov's model [26]. Function (V) is given in (6), which can make the linear screening methods fail. The data of predictors are generated as $\mathbf{X}_1, \ldots, \mathbf{X}_n$ identically and independently distributed from $U(0,1)^p$. Combinations of $(n, p, p_0)$ in our simulations can be found in Table 1.

**Table 1.** Coverage rates in interpolation.

| | Function (I) | | | | | |
|---|---|---|---|---|---|---|
| | $n = 100,$ $p_0 = 5$ | $p = 150$ $p_0 = 10$ | $n = 200,$ $p_0 = 5$ | $p = 500$ $p_0 = 10$ | $n = 400,$ $p_0 = 5$ | $p = 2000$ $p_0 = 10$ |
| DC-SIS | 0.298 | 0.004 | 0.423 | 0.012 | 0.596 | 0.034 |
| MDC-SIS | 0.328 | 0.003 | 0.461 | 0.018 | 0.669 | 0.039 |
| LSLB | 0.998 | 0.401 | 1.000 | 0.656 | 1.000 | 0.930 |
| LSQB | 1.000 | 0.984 | 1.000 | 1.000 | 1.000 | 1.000 |
| ensemble ($\delta = 0.6$) | 1.000 | 0.984 | 1.000 | 1.000 | 1.000 | 1.000 |
| ensemble ($\delta = 0.7$) | 1.000 | 0.984 | 1.000 | 1.000 | 1.000 | 1.000 |
| ensemble ($\delta = 0.8$) | 1.000 | 0.984 | 1.000 | 1.000 | 1.000 | 1.000 |
| | Function (II) | | | | | |
| | $n = 100,$ $p_0 = 5$ | $p = 150$ $p_0 = 10$ | $n = 200,$ $p_0 = 5$ | $p = 500$ $p_0 = 10$ | $n = 400,$ $p_0 = 5$ | $p = 2000$ $p_0 = 10$ |
| DC-SIS | 0.979 | 0.175 | 0.999 | 0.892 | 1.000 | 1.000 |
| MDC-SIS | 0.981 | 0.231 | 1.000 | 0.951 | 1.000 | 1.000 |
| LSLB | 1.000 | 0.871 | 1.000 | 1.000 | 1.000 | 1.000 |
| LSQB | 0.897 | 0.161 | 0.986 | 0.817 | 1.000 | 1.000 |
| ensemble ($\delta = 0.6$) | 1.000 | 0.872 | 1.000 | 1.000 | 1.000 | 1.000 |
| ensemble ($\delta = 0.7$) | 0.999 | 0.872 | 1.000 | 1.000 | 1.000 | 1.000 |
| ensemble ($\delta = 0.8$) | 0.997 | 0.855 | 1.000 | 1.000 | 1.000 | 1.000 |

**Table 1.** *Cont.*

| | Function (III) | | | | | |
|---|---|---|---|---|---|---|
| | $n = 100,$ $p_0 = 5$ | $p = 150$ $p_0 = 10$ | $n = 200,$ $p_0 = 5$ | $p = 500$ $p_0 = 10$ | $n = 400,$ $p_0 = 5$ | $p = 2000$ $p_0 = 10$ |
| DC-SIS | 0.981 | 0.296 | 1.000 | 0.933 | 1.000 | 1.000 |
| MDC-SIS | 0.988 | 0.372 | 1.000 | 0.962 | 1.000 | 1.000 |
| LSLB | 1.000 | 0.999 | 1.000 | 1.000 | 1.000 | 1.000 |
| LSQB | 1.000 | 1.000 | 1.000 | 1.000 | 1.000 | 1.000 |
| ensemble ($\delta = 0.6$) | 1.000 | 1.000 | 1.000 | 1.000 | 1.000 | 1.000 |
| ensemble ($\delta = 0.7$) | 1.000 | 1.000 | 1.000 | 1.000 | 1.000 | 1.000 |
| ensemble ($\delta = 0.8$) | 1.000 | 1.000 | 1.000 | 1.000 | 1.000 | 1.000 |
| | Function (IV) | | | | | |
| | $n = 100,$ $p_0 = 5$ | $p = 150$ $p_0 = 10$ | $n = 200,$ $p_0 = 5$ | $p = 500$ $p_0 = 10$ | $n = 400,$ $p_0 = 5$ | $p = 2000$ $p_0 = 10$ |
| DC-SIS | 0.277 | 0.003 | 0.420 | 0.026 | 0.537 | 0.042 |
| MDC-SIS | 0.286 | 0.004 | 0.411 | 0.013 | 0.621 | 0.055 |
| LSLB | 0.829 | 0.158 | 0.976 | 0.354 | 1.000 | 0.541 |
| LSQB | 0.860 | 0.134 | 0.981 | 0.320 | 1.000 | 0.470 |
| ensemble ($\delta = 0.6$) | 0.841 | 0.158 | 0.974 | 0.354 | 1.000 | 0.541 |
| ensemble ($\delta = 0.7$) | 0.841 | 0.158 | 0.974 | 0.354 | 1.000 | 0.541 |
| ensemble ($\delta = 0.8$) | 0.841 | 0.158 | 0.972 | 0.351 | 1.000 | 0.540 |
| | Function (V) | | | | | |
| | $n = 100,$ $p_0 = 3$ | $p = 100$ $p_0 = 5$ | $n = 200,$ $p_0 = 3$ | $p = 300$ $p_0 = 5$ | $n = 400,$ $p_0 = 3$ | $p = 800$ $p_0 = 5$ |
| DC-SIS | 0.775 | 0.155 | 0.978 | 0.573 | 1.000 | 0.998 |
| MDC-SIS | 0.045 | 0.003 | 0.016 | 0.002 | 0.005 | 0.000 |
| LSLB | 0.009 | 0.000 | 0.009 | 0.001 | 0.002 | 0.000 |
| LSQB | 0.012 | 0.002 | 0.014 | 0.001 | 0.001 | 0.000 |
| ensemble ($\delta = 0.6$) | 0.747 | 0.151 | 0.980 | 0.567 | 1.000 | 0.996 |
| ensemble ($\delta = 0.7$) | 0.772 | 0.155 | 0.982 | 0.573 | 1.000 | 0.998 |
| ensemble ($\delta = 0.8$) | 0.775 | 0.155 | 0.982 | 0.573 | 1.000 | 0.998 |

Fix $M = [n/\log(n)]$ suggested by [4]. Five methods are compared, including two SIS-type methods, two linear screening methods, and the proposed ensemble method. The two SIS-type methods are Li, Zhong, and Zhu [9]'s DC-SIS and Shao and Zhang [10]'s martingale difference correlation (MDC)-SIS method. The two linear screening methods, denoted by linear screening with linear basis (LSLB) and linear screening with quadratic basis (LSQB), respectively, adopt the linear and quadratic basis functions and use the lasso method, conducted with the Matlab package glmnet [27], to screen the important features. We use ten-fold cross-validation to specify the tuning parameter in lasso and only keep the variables with the largest $M$ absolute values of coefficients if the number of selected features is larger than $M$. In our method, three values of the threshold $\delta$, 0.6, 0.7, and 0.8, are used. The values of $\delta$ are selected by experience.

The coverage rates that the selected subset include for the true submodel over 1000 repetitions are given in Table 1. It can be seen from the table that MDC-SIS is slightly better than DC-SIS for Functions (I)–(IV) but performs poorly for Function (V). LSLB and LSQB have clear differences for some cases, which indicates that the selection of basis influences the performance of the linear screening methods. Although the linear screening methods have better overall performance than the SIS-type methods, they do not work for Function (V), which is pointed out in Example 4. For the proposed ensemble method, we find that

the value of $\delta$ does not heavily influence its performance. In addition, ensemble screening overcomes the limitations of the SIS-type and linear screening methods and performs very close to the best method among DC-SIS, MDC-SIS, LSLB, and LSQB for all the cases.

As mentioned in Section 3.1, Azadkia and Chatterjee proposed a forward variable selection method for nonparametric models via a measure of conditional dependence [20]. This method usually yields very sparse selection results and has relatively good performance for $p < n$ problems in our numerical experience. Note that we can use it as a screening method when conducting it until $M$ variables have been selected. However, as a screening method, this forward method performs poorly when $p > n$. For example, for $(n, p, p_0) = (100, 150, 5)$ and $M = [n / \log(n)]$, its coverage rates of correct screening over 1000 repetitions for Functions (I)–(III) are only 0.048, 0.116, and 0.019, respectively.

We also conduct simulations for regression and classification problems with the test functions and report the results in the Supplementary Materials. It can be seen from the regression and classification results that the main findings are similar to those in the interpolation cases.

The dataset in this subsection contains $N = 9568$ data points collected from a Combined Cycle Power Plant (CCPP) over 6 years (2006–2011) [28], when the power plant was set to work with a full load. Features consist of hourly average ambient variables Temperature, Ambient Pressure, Relative Humidity, and Exhaust Vacuum to predict the net hourly electrical energy output of the plant.

We first standardize the data of the four features into $[0, 1]^4$ via the linear transformation $x_{ij} \leftarrow (x_{ij} - \min_{k=1,\dots,N} x_{kj}) / (\max_{k=1,\dots,N} x_{kj} - \min_{k=1,\dots,N} x_{kj})$ for $i = 1, \dots, N$, $j = 1, 2, 3, 4$. Then, we randomly draw a subsample $\{x_{ij}\}_{i=1,\dots,200,j=1,2,3,4}$ of 200 data points from the whole dataset, and add 296 noisy variables to construct the test dataset of $n = 200$ and $p = 300$ to evaluate the screening methods. To generate the noisy features, we draw a number $J$ among $\{1, 2, 3, 4\}$ with equal probabilities, and let $x_{ij} = \max\{\min\{x_{iJ} + z, 1\}, 0\}$, $i = 1, \dots, n$, $j = 5, \dots, p$, where $z \sim N(0.5, 0.4^2)$. It can be seen that the noisy variables are correlated with the true features. We repeat the process of generating the test datasets 1000 times and report the coverage rates for which the screening methods correctly select the four true features with different $M$ in Figure 1.

It can be seen from the results that the two SIS-type methods perform very similarly. The two linear screening methods perform poorly. However, with their help, the proposed method outperforms DC-SIS and MDC-SIS.

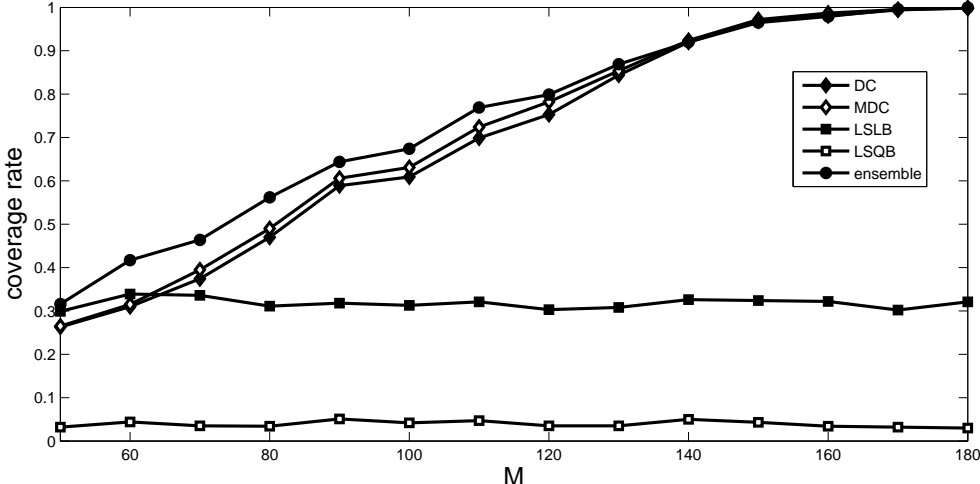

**Figure 1.** Coverage rates in Section 5.2.

*5.2. Computer Tomography Images*

The dataset in this subsection was retrieved from a set of 583 computer tomography images from a person. Each computer tomography slice is described by two histograms

in polar space. The first histogram has 240 components describing the location of bone structures in the image. The second histogram has 144 components, describing the location of air inclusions inside the body. Both histograms are concatenated to form the final feature vector. The response variable is the relative location of an image on the axis, which was constructed by manually annotating up to 10 different distinct landmarks in each computer tomography volume with a known location. A more detailed description of the dataset can be found in [29]. Among those 583 images, 350 of them are randomly selected as the training sample, and the remaining 233 are set as the test sample. Therefore, the sample size *n* is 350, less than the number of variables, *p* = 384, in the feature vector.

Based on the training sample, we conduct the screening methods to reduce the number of predictors from *p* to *M*. Like in [20], we then use the random forest (RF) algorithm [30], conducted by the `Matlab` function `TreeBagger` with 500 decision trees in the ensemble, to predict the response on the test sample. The mean test errors of these methods over 100 replications are presented in Figure 2. We also show the test errors of random forest without the screening step.

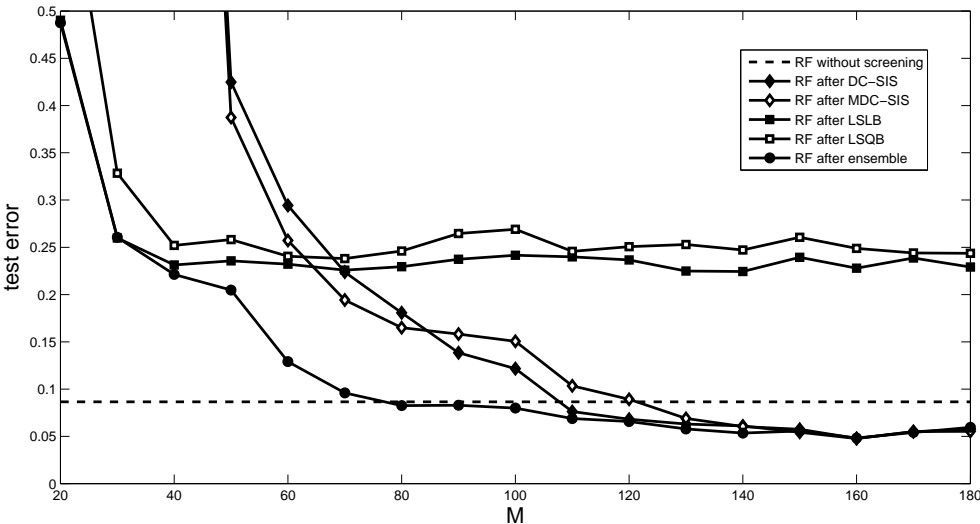

**Figure 2.** Test errors in Section 5.2.

It can be seen from the results that, for a small *M*, the two SIS-type methods perform very poorly. The proposed method always yields the smallest test errors among the five methods. In particular, with about 70 variables selected by our method, the test error is very close to that with all 384 variables, while DC-SIS and MDC-SIS need about 110 variables to achieve a similar result.

### 5.3. Voice Rehabilitation

The dataset from a study on objective automatic assessment of rehabilitative speech treatment in Parkinson's disease contains 126 data points with 310 predictors. The predictors are all continuous values, and the response is binary. Detailed information on the predictors and response can be found in [31]. Among those 126 data, we randomly selected 100 of them as the training sample, and the remaining 26 were set as the test sample.

In addition to the methods compared in the previous subsections, we add Cui, Li, and Zhong [32]'s MV-SIS method, which is a model-free feature screening approach for high-dimensional classification problems. Similarly, we conducted the six screening methods to reduce the number of predictors from 310 to 15. Then the random forest algorithm was used to predict the response on the test sample. The mean test errors of these methods over 100 replications are presented in Table 2. We also show the test errors of random forest without the screening step. From Table 2, we find that the two linear screening methods and the ensemble method effectively reduce the dimensionality: their test errors are less than that from random forest without the screening step. In addition, the performance

of the ensemble screening method is very close to the best method, LSQB, among these screening methods. This point is consistent with our findings in simulations with test functions in the Supplementary Materials.

**Table 2.** Test errors of RF in the real data example.

|  | Full Model | DC-SIS | MDC-SIS | MV-SIS | LSLB | LSQB | Ensemble |
|---|---|---|---|---|---|---|---|
| Mean | 0.163 | 0.232 | 0.231 | 0.361 | 0.142 | 0.136 | 0.138 |
| Standard deviation | 0.069 | 0.083 | 0.086 | 0.109 | 0.058 | 0.057 | 0.053 |

## 6. Concluding Remarks

For the screening problem in nonparametric settings, SIS-type methods have some limitations. The linear screening methods, from the perspective of applied statistics or data analytics, are alternatives. A natural idea is to combine the two classes of methods to overcome their shortcomings. The ensemble method proposed in this paper follows this line. Numerical investigations indicate that it performs stably even when existing methods fail. For highly correlated predictors, the threshold $\delta$ in ensemble screening can be set as 0.8 or larger, while for other cases, $\delta = 0.7$ seems to be a good choice. This paper focuses on continuous and binary responses. The method proposed here can be applied to other types of responses, such as counting or multivariate responses, without any extra difficulties.

**Supplementary Materials:** The following supporting information can be downloaded at: https://www.mdpi.com/article/10.3390/math11020362/s1, Table S1: Coverage rates in regression; Table S2: Coverage rates in classification, and the reference in Supplementary Materials was cited in Ref. [32].

**Author Contributions:** Methodology, X.W. and S.X.; Validation, S.X.; Formal analysis, X.W.; Investigation, W.M.; Data curation, X.W.; Writing—original draft, X.W.; Writing—review & editing, W.M.; Supervision, W.M.; Project administration, W.M.; Funding acquisition, S.X. All authors have read and agreed to the published version of the manuscript.

**Funding:** This work is supported by the National Key R&D Program of China (Grant nos. 2021YFA1000300, 2021YFA1000301, and 2021YFA1000303) and the National Natural Science Foundation of China (Grant no. 12171462).

**Data Availability Statement:** Not applicable.

**Acknowledgments:** We thank the editors and referees for constructive comments which lead to a significant improvement of this article. This work is supported by the National Key R&D Program of China (Grant nos. 2021YFA1000300, 2021YFA1000301, and 2021YFA1000303) and the National Natural Science Foundation of China (Grant no. 12171462).

**Conflicts of Interest:** The authors declare no conflicts of interest.

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
