# Peer review of "An Ensemble Method for Feature Screening"

_mathematics, doi:10.3390/math11020362_

Round 1

Reviewer 1 Report

The authors introduce a novel method of feature selection of high-dimensional nonparametric models by combining many existing screening methods. Specifically, ensemble screening is based on the R-squared principle. The method is trained on several scenarios and uses various tuning parameters. Additionally, empirical real data sets are used for illustrative purposes. Finally, theoretical justifications for the proposed method are proven and they seem sound.

Author Response

please see the attached pdf file.

Reviewer 2 Report

My comments are attached.

Author Response

please see the attached pdf file.

Reviewer 3 Report

The contribution of the current manuscript is not sufficient for publication in this Journal. Thus,  I regret to inform you that I have decided against publishing your manuscript.

Author Response

please see the attached pdf file.

Reviewer 4 Report

   By: Xi Wu , Shifeng Xiong , Weiyan Mu

Submitted to: Mathematics

Report     12/18/2022

Major Comments: 

The authors point out some limitations of existing screening methods, then porpose

a nonparametric screening procedure for feature selection with high-dimensional

data. Simulation studies are conducted to evaluate the performance of the proposed

method.

* Feature screening is an extensively studied topic, the review of related methods 

  should be more broad. The authors should discuss the relationships of their method

  to, such as the methods of Bai and Saranadasa (1996) and Chen and Qin (2010).

* For the distance covariance dcov(U,V) in page 6, please give a discussion of its

  relationship between the energy distance in Szekely (1985) and Szekely et al. (2004). 

* In equation (5), page 9, \beta is a p-dimensional vector (p>n), the authors should 

  explain how the inverse is computable?

* The Lasso is used to select important features (page 11-12). The authors should

  give a discussion of the issues of post-lasso inference.

Minor Comments.

* The authors pointed out several issues with the existing methods in Section 3. It 

  is not easy to see where the issues are. Please put them in a easy-to-see format,

  such as Propositions, Facts,...

Author Response

please see the attached pdf file.

Round 2

Reviewer 4 Report

I'm satisfied with the revision. After some minor edits in English,

the manuscript is now acceptable for publication. 

Author Response

please see the pdf file. 
